# Determining jumping performance from a single body-worn accelerometer using machine learning

**Mark G. E. White** [1,2]***, Neil E. Bezodis**[1]**, Jonathon Neville**[3]**, Huw Summers**[2]**, Paul Rees**[2]

**1** Applied Sports, Technology, Exercise and Medicine Research Centre, Swansea University, Swansea, United Kingdom, **2** Department of Biomedical Engineering, Swansea University, Swansea, United Kingdom, **3** Sport Performance Research Institute New Zealand, Auckland University of Technology, Auckland, New Zealand

* markgewhite@gmail.com

**Data Availability Statement:** All data files are available for download from here: DOI 10.17605/OSF.IO/K4WRB Matlab code is also available to process this data.

## Abstract

External peak power in the countermovement jump is frequently used to monitor athlete training. The gold standard method uses force platforms, but they are unsuitable for field-based testing. However, alternatives based on jump flight time or Newtonian methods applied to inertial sensor data have not been sufficiently accurate for athlete monitoring. Instead, we developed a machine learning model based on characteristic features (functional principal components) extracted from a single body-worn accelerometer. Data were collected from 69 male and female athletes at recreational, club or national levels, who performed 696 jumps in total. We considered vertical countermovement jumps (with and without arm swing), sensor anatomical locations, machine learning models and whether to use resultant or triaxial signals. Using a novel surrogate model optimisation procedure, we obtained the lowest errors with a support vector machine when using the resultant signal from a lower back sensor in jumps without arm swing. This model had a peak power RMSE of 2.3 W·kg$^{-1}$ (5.1% of the mean), estimated using nested cross validation and supported by an independent holdout test (2.0 W·kg$^{-1}$). This error is lower than in previous studies, although it is not yet sufficiently accurate for a field-based method. Our results demonstrate that functional data representations work well in machine learning by reducing model complexity in applications where signals are aligned in time. Our optimisation procedure also was shown to be robust can be used in wider applications with low-cost, noisy objective functions.

## Introduction

The ability to generate high levels of neuromuscular power is a critical aspect of sports performance [1–3]. It is strongly correlated with sprint acceleration [4–8] and serves as an indicator of overtraining or fatigue [9–12]. Accordingly, peak external power in the countermovement jump (CMJ) is monitored frequently in many professional athletes [13,14]. Reports of peak

**Funding:** The authors received no specific funding for this work.

**Competing interests:** The authors have declared that no competing interests exist.

power reductions typically range from 3.5% for a training protocol [10] to 12.4% for Australian rules football players 24 hours after a match [9]. The gains in peak power from training can be more substantial with average increases over three years of 13% and 46% reported for male Australian Rules football players and female collegiate gymnasts, respectively [15,16].

Jump testing has practical advantages as it can be administered quickly without residual fatigue, typically before a training session [13,14]. However, the gold-standard method relies on force platforms, which are cumbersome, expensive and unsuited to field-based testing [17]. Many coaches prefer to measure jump height instead of peak power as it requires minimal setup time and equipment [14]. However, although it has some factorial validity [18], jump height is a distinct measure from peak power ($r = 0.93$) [19]. Whereas jump height depends on the cumulative work done (impulse converts to take-off velocity through the conservation of momentum), power is an instantaneous measure reflecting the ability of the locomotor apparatus to perform rapid movements.

Formulae have been proposed for predicting peak power from jump height and body mass, with errors of 247–562 W, equivalent to 6.0–16.5% [20–25]. When those same equations were tested independently, their errors ranged more widely from 3.8–25.3% [24–27]. In recent years, researchers have investigated the use of body-worn inertial sensors to estimate instantaneous power from the body's vertical acceleration and velocity. The peak power estimates, however, were similarly inaccurate with errors of 10.7–21.2% [28–30]. These Newtonian approaches are highly sensitive to the small errors that arise from corrections needed to the sensor's changing orientation [31–33]. Moreover, even without such errors, the computed peak power would not be the same as the true external power because the sensor does not follow the trajectory of the body's centre of mass [34].

Rather than computing peak power directly from the signal, a machine learning approach may be more successful by relating patterns in the data to the performance outcome. Machine learning (ML) models and deep neural networks have been used to predict discrete performance measures derived from the ground reaction force (VGRF), such as peak force or loading rate [35–40]. The ML models in these studies performed at least as well as the neural networks, but without needing high data volumes that can be challenging to obtain, especially in the study of human movement. Different techniques for extracting characteristic features from the data have ranged from devising bespoke metrics to collecting generic statistical measures or employing dimensional reduction techniques such as Principal Component Analysis (PCA). Features based on functional principal components (FPCs) were considerably more accurate than expert-determined discrete measures when predicting jump height from VGRF data [41,42]. Indeed, Functional Principal Component Analysis (FPCA) has been applied to a diverse range of applications in biomechanics. Studies have reported strong associations between FPCs and various performance or injury risk measures in sports, including rowing, swimming, weightlifting, race walking and jumping [41,43–51]. These applications analysed the FPC scores using descriptive statistics, *t*-tests, ANOVA, discriminant analysis or a simple regression model to address their research questions. However, more sophisticated ML models in conjunction with FPCA have not yet been investigated.

When developing an ML model, its parameters need to be tuned through an optimisation procedure, typically using cross validation. However, if model selection is not made independently of model evaluation, then the model selection bias leads to an under-estimation of the model's generalised predictive error [52–55]. Nested cross validation (NCV), also known as double cross validation, overcomes this problem by enforcing the separation between model selection and evaluation, yielding unbiased error estimates [55–58]. It allows different K-fold cross validation (CV) designs to be used for model selection and evaluation, which have

distinct requirements [59]. However, despite its advantages NCV is rarely used in machine learning studies, not least in biomechanics.

This paper presents new models based on functional principal components for predicting peak power in the CMJ from body-worn sensor data. We focus on a single sensor solution for practical reasons as athletes often wear a single inertial measurement unit (IMU) in team sports. Our aim was to produce a model with a predictive error smaller than a typical athlete's inter-day variability. In order to obtain a threshold value for this inter-day variability *a priori*, we averaged the reported inter-day variability in trained athletes across three studies [10,60,61], obtaining a target error level of 3.4%. We modified existing techniques to develop a novel and rigorous optimisation procedure within a nested cross validation framework [62–64]. The optimisation concerned parameters for data preprocessing and the model itself, thereby encompassing the whole modelling procedure [59]. We used this procedure to answer the following research questions: (1) How accurately can peak external power be determined during a CMJ using an ML model based on body-worn accelerometer data? (2) Which of the anatomical locations considered is best for the sensor? (3) How should the signal data be processed?

## Materials and methods

### Data collection

We recruited 69 healthy participants (45 males, 24 females: body mass 73.1 ± 13.1 kg (mean ± SD); height 1.74 ± 0.10 m; age 21.6 ± 1.5 years) who gave their written informed consent. The study was approved by the Research Ethics and Governance Committee of Swansea University's College of Engineering. All the participants played a sport, either at recreational (15), club (43) or national (11) level, except for four who trained regularly in the gym. The most frequent sports were football (10), volleyball (7), netball (5), rugby union (5) and rowing (5). The participants each performed either 8 or 16 maximal effort CMJs, divided equally between jumps with arm swing ($CMJ_A$) and those without ($CMJ_{NA}$), where hands were placed on hips. Most participants (55) completed 8 jumps as they also performed 8 broad jumps as part of a wider research project. The order of jumps was randomised to minimise potential learning and fatigue effects. The participants were given one minute's rest between each jump. All jumps were performed on two portable 400 × 600 mm force platforms (9260AA, Kistler, Winterthur, Switzerland), which recorded the vertical component of the ground reaction force at a sampling frequency of 1000 Hz. For convenience, all abbreviations used in this paper are listed in Table 1.

The unfiltered VGRF data, summed from both platforms, with body weight (BW) subtracted, gave the net force. The resulting acceleration (i.e. net force/mass) was integrated using the trapezoidal rule to obtain the vertical velocity. The product of velocity and VGRF gave the instantaneous power, from which the maximum value, normalised to body mass, gave the criterion value for peak power in the models below ($W \cdot kg^{-1}$). Jump initiation, the start point for the integration procedure, was identified using a two-step procedure adapted from [65]. The jump was detected initially where VGRF deviated by more than 8% BW, yet the movement must have begun earlier. Rather than using a fixed 30 ms backwards offset [65], the offset depended on where the VGRF deviation had exceeded 1% BW immediately before reaching the 8% threshold.

Delsys Trigno sensors (Delsys Inc., Natick, MA, USA) were attached over the L4 vertebra on the lower back (LB sensor), the C7 vertebra on the upper back (UB sensor), and the lower anterior medial aspect of the tibias (LS/RS sensors), three anatomical positions commonly used in field-based testing [66] (Fig 1). They were attached directly to the skin using double-

**Table 1. Summary of abbreviations.**

| Acronym | Definition |
| --- | --- |
| AM | Accelerometer Model |
| ANOVA | Analysis of Variance |
| BW | Body Weight |
| CMJ | Countermovement Jump |
| CV | Cross Validation |
| FPC | Functional Principal Component |
| FPCA | Functional Principal Component Analysis |
| GPR | Gaussian Process Regression |
| GPS | Global Positioning System |
| IMU | Inertial Measurement Unit. |
| LB | Lower Back |
| LR | Linear Regression |
| LS | Left Shank |
| ML | Machine Learning |
| NCV | Nested Cross Validation |
| PCA | Principal Component Analysis. |
| PSO | Particle Swarm Optimisation. |
| RMSE | Root Mean Squared Error. |
| RS | Right Shank |
| SM | Surrogate Model). |
| SVM | Support Vector Machine |
| UB | Upper Back |
| VGRF | Vertical Ground Reaction |

sided surgical tape and held firmly in place by an elastic adhesive bandage to minimise soft-tissue movement [67,68]. The sensors transmitted the analogue triaxial accelerations (±9 g) for each jump to a receiving station connected to a computer. Vicon Nexus v2.5 software (Vicon, Oxford, UK) sampled the analogue accelerometer data at 250 Hz and synchronised it with the VGRF data. Although the sensors could digitally sample the measurements, the analogue form

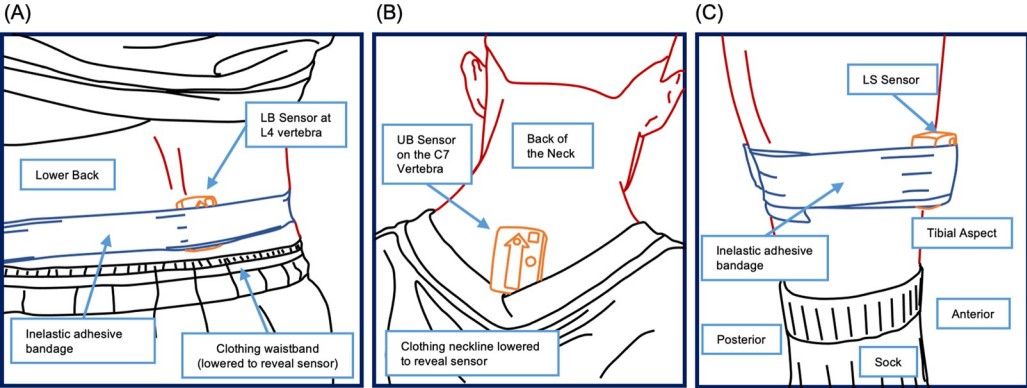

**Fig 1. Illustration showing the anatomical position of the inertial sensors.** (A) Lower back (LB) sensor attached with double-side tape and held in place with an inelastic adhesive bandage wrapped around the waist; (B) Upper back (UB) sensor attached only with double-sided tape; (C) Left shank (LS) sensor also attached with the same tape and held firmly in place by an adhesive bandage wrapped around the leg. The right shank (RS) sensor (not shown) was attached in the same way.

made the direct synchronisation of accelerometer and VGRF data possible. The sensors were calibrated following the manufacturer's instructions by placing them in six stationary, orthogonal orientations.

## Data processing

Data from 696 CMJs were recorded, although four jumps had to be discarded owing to an issue with the accelerometer data. The data (VGRF and accelerometer time series) from 60 participants were assigned to a training/validation data set (548 jumps), while data from the remaining 9 participants (randomly chosen) were placed in an independent holdout test set (144 jumps). All bodyweight-normalised VGRF time series were padded to a standard length, the longest time series. A series of 1's was inserted (i.e. equal to body weight), as required, at the start of the time series to mimic quiet standing before the jump and at the end to reflect the standing position regained after the landing. A similar operation was performed on the accelerometer signal, inserting values equal to the mean acceleration recorded at the start in quiet standing. However, the optimal model may not require the full-length time series, so data were extracted from a time window beginning at a specified time before take-off ($t_{pre}$) and ending at a time after take-off ($t_{post}$). These two parameters were allowed to vary in steps of 0.1 s. The take-off time for the accelerometer data was identified where the VGRF dropped below 10 N for the first time [69]. The landing time was when VGRF subsequently rose above 10 N. The flight time was the difference between the take-off and landing times. All processing, unless otherwise stated, was performed in MATLAB R2021a (MathWorks, Natick, MA, USA). The code is available from GitHub: https://github.com/markgewhite/accModel.

The accelerometer signals were padded to the same duration as the VGRF data with the mean acceleration vector over the first 0.5 s when the participant stood still before the jump. The signals were then converted into smooth, continuous functions using b-splines [42]. The number of basis functions was defined indirectly as a density ($\rho$, b-splines per unit time) to make it independent of the time window parameters. The basis function specification ($\Phi$) incorporated the basis order and the penalty order for the roughness penalty. The basis ranged from $4^{th}$ order (cubic) to $6^{th}$ order to offer more flexibility and greater b-spline overlap. The accelerometer signal was smoothed either by penalising high curvature ($2^{nd}$ order derivative) or by the rate of change of curvature ($3^{rd}$ order derivative), which would permit abrupt acceleration changes, such as preserving the amplitude of the high acceleration peak on landing. Using a single categorical parameter, $\Phi$, rather than having two parameters (basis and penalty order) reduced the parameter space dimensionality and provided a list of valid combinations (see Search Range in Table 2). $N_C$ specified the number of retained Functional Principal Components (FPCs) [42]. No varimax rotation was used in order to preserve the FPCs' independence and avoid multicollinearity. These procedures were applied to the accelerometer data from all four sensors, providing separate data sets for the ML models. All the parameters are summarised in Table 2.

## Modelling procedures

Three common machine learning models were considered: regularised linear regression (LR), support vector machine (SVM) and Gaussian process regression (GPR). The regression models' predictor variables were the accelerometer FPC scores, and the outcome variable was the peak external power computed from the VGRF data. The model hyperparameters and the data processing parameters were determined through the optimisation procedure described below.

**Nested cross validation.** The optimisation procedure was run within an NCV framework to produce unbiased estimates of the model's generalised predictive error [54–57,70]. The data were first partitioned at the participant level with a 10-fold design for the outer loop. Jumps

**Table 2. Data processing parameters with their respective ranges.**

| Model | Parameter | Description | Type † | Search Range ‡ | Optimisation Bounds § |
|---|---|---|---|---|---|
| All | $t_{pre}$ | Time before take-off | I | $[-35, \ldots, 5] \times 100$ ms | [0.51, 30.49] |
| | $t_{post}$ | Time after take-off | I | $[-5, \ldots, 35] \times 100$ ms | [0.51, 30.49] |
| | $\rho$ | Density of basis functions per second | R | $[2 \ldots 22]$ | [4, 20] |
| | $\Phi$ | Basis function (order and penalty derivative) | C | {4-2, 5-2, 5-3, 6-2, 6-3, 6-4} * | [0.51, 6.49] |
| | $\lambda$ | Roughness penalty | R | $[-12 \ldots 12]$ | [-10, 10] |
| | $N_C$ | No. retained FPCs | I | $[1, \ldots, 35]$ | [3.51, 30.49] |
| | $Z$ | Standardisation ⌐ | C | {No, Yes} | [0.51, 2.49] |
| LR | $R$ | Regularisation Method | C | {Ridge, Lasso} | [0.51, 2.49] |
| | $S$ | Solver Method | C | {SVM, Least Squares} | [0.51, 2.49] |
| | $\lambda_{LR}$ | Regularisation Parameter | R | $[-12 \ldots 12]$ | [-10, 10] |
| SVM | $K_{SVM}$ | Kernel | C | {Gaussian, Linear, Polynomial} | [0.51, 3.49] |
| | BC | Box Constraint | R | $[-7 \ldots 9]$ | [-6, 8] |
| | KS | Kernel Scale | R | $[-7 \ldots 9]$ | [-6, 8] |
| | $\varepsilon$ | | R | $[-5 \ldots 3]$ | [-4, 2] |
| GPR | $B$ | Basis | C | {None, Constant, Linear, Pure Quadratic} | [0.51, 4.49] |
| | $K_{GPR}$ | Kernel | C | {Exponential, Squared Exponential, Matérn 3/2, Matérn 5/2, Rational Quadratic} | [0.51, 5.49] |
| | $\sigma$ | Noise | R | $[-5 \ldots 3]$ | [-4, 2] |

† Parameter type: C = Categorical; I = Integer; R = Real (continuous, $\log_{10}$ transformed).

‡ Random search range has a broader range to gather data outside the optimisation bounds to prevent boundary effects–see text. Further constraints are imposed by the surrogate model.

§ Particle Swarm Optimisation uses real parameters, so the bounds extend 0.50 below and 0.49 above for categorical (indexed) and integer parameters so when rounded, there is no bias at lower and upper limits.

* Basis encoding = <*basis order*>-<*penalty order*>, defines valid combinations.

⌐ Standardise the predictors and outcome variables as Z scores during fitting.

from 54 of the 60 participants were assigned randomly to a training set for each iteration, while the remaining 6 participants were placed in a validation set (Fig 2A). The 10-fold partitioning was repeated (2 × 10 outer loop iterations) to reduce uncertainty in the predictive error estimate [57,71,72]. A 10-fold design was recommended for model evaluation as validation error estimates have low bias [59,73] and to provide a large proportion of the data for the model selection [55]. For each outer iteration, model selection was performed on the outer training set using 2-fold CV. The inner training and validation sets both comprised data from 26 participants (Fig 2B). Two-fold CV provides a large validation set to increase the likelihood of selecting the best regression model [74,75]. The best model that emerged from the inner loop was evaluated on the outer validation set. Since this data had been kept separate, the validation RMSE was an independent test of the whole modelling procedure.

**Objective function.** The accelerometer model function (AM) performed all aspects of the modelling process and served as the objective function, returning the 2-fold validation RMSE (loss) for the outer training data set. It carried out time series padding, functional smoothing, data partitioning and the CV inner loop, including FPCA and model fitting, prediction and validation error calculations. The AM defined FPCs based on the inner training partition alone and used them to compute the FPC scores for both the inner training and validation sets. It penalised invalid parameter combinations by returning a high loss (10 W·kg⁻¹). Invalid parameter combinations arose when there were insufficient basis functions for the number of

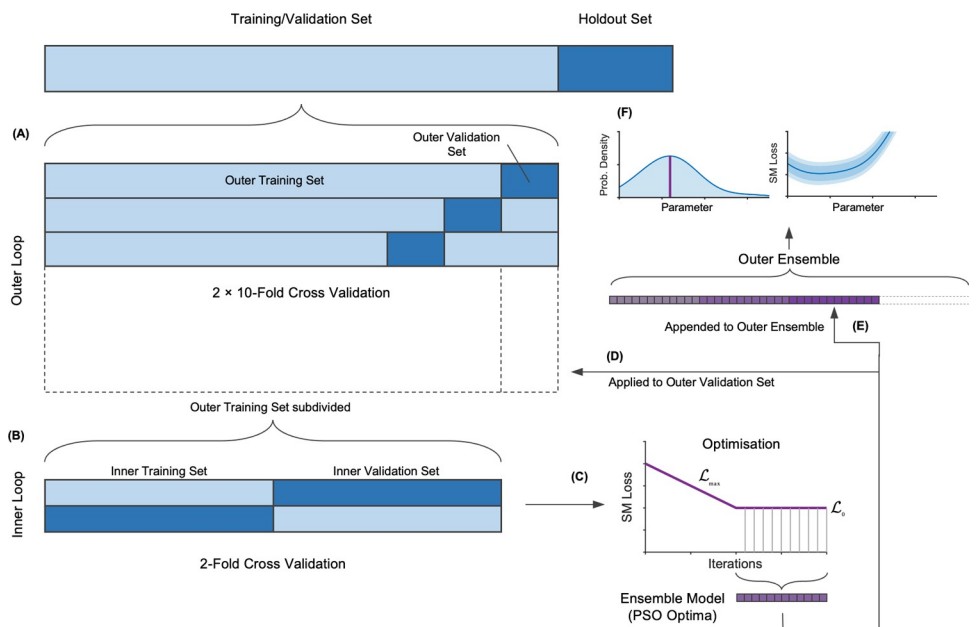

**Fig 2. Schematic design of the nested cross validation and optimisation procedure.** (A) The training/validation set is partitioned 10-fold for the outer loop, and then (B) each training outer set is re-partitioned 2-fold for the inner loop. (C) Optimisation works with the observations from the inner loop to determine an ensemble model based on the series of optimal parameters determined by Particle Swarm Optimisation. (D) The ensemble model is then evaluated on the outer training set. (E) The process repeats for each outer fold, adding to the series of optimal parameters used to determine the outer ensemble model. (F) This yields parameter distribution and partial plots. It also produces the final ensemble model that may be applied to the holdout data.

FPCs required or if the resulting FPC-score training matrix did not have full column rank. Losses were also capped at 10 W·kg$^{-1}$ to prevent very occasional extreme losses from destabilising the optimisation.

**Surrogate model.** A Bayesian approach is needed to accommodate the objective function's stochastic behaviour arising from the CV subsampling variance. Many AM observations were required given the high-dimensional parameter space, but we found Bayesian optimisation became prohibitively expensive as more observations were added. Instead, we adapted a random search procedure [76] with a low overhead so it was tractable to make hundreds of observations. A surrogate model (SM) was fitted to the observations based on a Gaussian Process (GP), thus retaining the Bayesian approach [77]. The SM had the same specification used by MATLAB for its *bayesopt* optimiser: an anisotropic Matérn 5/2 kernel and a constant basis function with no predictor standardisation.

**Constrained random search.** The random search made 400 observations in each optimisation that were constrained to regions of the parameter space where the SM predicted a low AM loss. The constraint was imposed according to the following probability function that governed whether a randomly generated point was accepted:

$$p = \begin{cases} 1, & \mathcal{L}_i \leq \mathcal{L}_{\max} \\ \exp\left[\dfrac{-(\mathcal{L}_i - \mathcal{L}_{\max})^2}{2\delta^2}\right], & \mathcal{L}_i > \mathcal{L}_{\max} \end{cases}$$

where

$$\mathcal{L}_{\max} = \mathcal{L}_0 + \alpha_j \mathrm{SD}(\mathcal{L}), \quad \delta = {}^1/_2 \mathrm{SD}(\mathcal{L})$$

$\mathcal{L}_i$ is the SM prediction for the $i$-th observation using the $j$-th surrogate model, such that $i \in \{1 \ldots 400\}$ and $j \in \{1 \ldots 20\}$. Hence, the SM was retrained every 20 observations. $\mathcal{L}_{max}$ is a progressively declining upper limit to a baseline loss, $\mathcal{L}_0$. $\delta$ is a measure of how likely points exceeding $\mathcal{L}_{max}$ will be accepted. $\alpha_j$ has a ramp profile to constrain the search such that it decreases linearly from 0.5 to 0 over the first half of the search and thereafter remains at zero (Fig 2C). Thus, the search initially surveys the parameter space when almost all points are accepted before tightly focusing its search in promising regions, relying almost entirely on the probability function. Often hundreds of candidate points could be rejected until one is accepted, but the overhead was minimal as the SM predictions were computed quickly (~0.0005 s vs 0.2 s for the AM).

**Optimisation.** The SM was deterministic so global optimisers other than those using Bayesian methods could be employed. We chose Particle Swarm Optimisation (PSO) as it has been used to good effect in previous model optimisation problems [78–80]. It was set up to use 100 particles with an objective tolerance of 0.001. Since PSO only works with continuous variables, it was necessary to index the categorical parameters (Table 2) and use an intermediate objective function that rounded the categorical and integer parameters to the nearest whole number. PSO was run after the SM was retrained on these indexed parameters (Fig 2C). The random search was wider than PSO parameter bounds in order to populate a border region to ensure the SM was well-defined at the periphery (Table 2, last column).

**Ensemble models.** Once the random search and the final PSO were complete, an ensemble model was selected based on maximum likelihood. For categorical parameters, the most frequently occurring category was chosen, and for numeric parameters, the value with the highest probability density. The values were drawn from the series of PSO-determined optimal models, taken from the second half of the search when $\alpha_j = 0$, provided the constraints were satisfied (Fig 2C). The ensemble model was trained on the outer training set and evaluated on the outer validation set (Fig 2D). The NCV predictive error estimate was the average outer validation RMSE. Since model selection yielded a different model for each outer fold, the same maximum likelihood procedure was used to determine the final ensemble model parameters for the whole data set. The procedure ran on an aggregated list of PSO-determined optimal models taken from all outer folds (Fig 2E). The ensemble AM was trained on the entire training/validation set and then evaluated on the holdout test set to provide a final independent test of the model (Fig 2F).

## Model analysis

**Statistical comparisons.** The whole modelling procedure above was run for each combination of model type (LR, SVM, GPR), sensor location (LB, UB, LS, RS) and jump type (CMJ$_{NA}$, CMJ$_A$). This analysis was performed on data sets based on the resultant or triaxial accelerometer signals to determine the best signal representation. The outer validation errors were compared between conditions (signal representation, model type, sensor location, jump type) using a two-way ANOVA with 960 observations (2 signal representations × 3 model types × 4 sensor locations × 2 jump types × 20 outer folds). The ANOVA model was another surrogate model predicting AM loss, which, although inferior to the GP model, allowed hypothesis testing. Effect sizes were based on semi-partial $\omega^2$, the proportion of the total variance (significance level 0.05) [81]. It was necessary to Winsorise all the data because a few outer validation errors for the SVM models were extremely large ($\gg 10$ W·kg$^{-1}$, including six $> 20$ W·kg$^{-1}$, three $> 40$ W·kg$^{-1}$), rendering otherwise significant effects undetectable. Accordingly, ten observations at opposite ends of the range were adjusted, equivalent to the 1st and 100th percentiles.

The statistical procedures were run in SAS Studio 3.8 (SAS Institute Inc., Cary, NC, USA) using *Proc GLM*. These procedures were bootstrapped (1000 replicas, stratified by condition) to obtain robust estimates because there was no homogeneity of variances at the model type level, according to Levene's test (no suitable transformations would suffice). The bootstrapped estimates are reported with 90% confidence intervals using the median for the central estimate and the 5th and 95th percentiles for the limits.

**Model refinement.** We selected the dataset with the lowest RMSE across the three model types for further refinement through repeated optimisations. For each model type, some parameter distributions indicated a strong preference for a certain optimal value. In addition, the associated SM partial plots generally showed an advantageous lower predicted loss. Where this was the case, the parameter was fixed at this value, removing it from the optimisation. We judged this subjectively as no satisfactory objective rules could be devised. In other cases, where there was no clear choice, specific values could not be excluded from the search range. Four rounds of optimisation were run for each model type, successively intensifying the search each time. The fourth and final optimal model was then applied to the holdout data set as an independent test.

## Results

The peak power computed from the VGRF data (criterion measure) was similar between the training/validation and holdout groups, with higher peak powers recorded in jumps with arm swing (Table 3).

The bootstrapped ANOVA reported an overall effect of $F_{(9,959)} = 24.7$ [19.3, 31.1], $p < 0.0001$ with total $\omega^2 = 0.190$ [0.155, 0.228]. The strongest effects on the outer validation error were made by model type and jump type, respectively, explaining 10.6% and 4.6% of the variance (Table 4). These two factors, and sensor location, were the only ones that were significant across the 90% confidence interval. Signal representation did not always reach significance as the bootstrapped interval for the *p*-value extended beyond 0.05. It explained less than 1% of the variance, as did the interaction between model type and jump type, the only significant interaction.

The distributions of the Winsorised outer validation errors, grouped by condition, are shown in Fig 3 (top row), revealing which levels within each condition yield more accurate models. Predictions of peak power in the CMJ$_{NA}$ are significantly more accurate in absolute terms than in the CMJ$_A$: 3.82 W·kg$^{-1}$ vs 4.62 W·kg$^{-1}$ (Fig 3A). However, relative to the data set's mean peak power the difference was less marked: 8.5% vs 9.0%. Using the LB sensor yielded more accurate models than when sensors were located elsewhere (Fig 3B), although this difference only reached significance compared to RS models. The errors of the UB, LS and RS sensor-based models were not significantly different from one another. Models based on the resultant accelerometer signal were marginally more accurate than those based on the tri-axial signal, but this difference was not significant (Fig 3C). The model types' errors were all significantly different from one another, with the GPR model being most accurate (Fig 3D).

**Table 3. Peak power (W·kg$^{-1}$) computed from VGRF data.**

|  |  | Mean ± SD | 10th– 90th Percentile | Min, Max |
|---|---|---|---|---|
| **Training / Validation data set** | CMJ$_{NA}$ | 45.0 ± 7.3 | 35.2–54.1 | 27.2, 63.6 |
|  | CMJ$_A$ | 51.5 ± 8.6 | 39.7–62.1 | 28.1, 72.5 |
| **Holdout data set** | CMJ$_{NA}$ | 47.6 ± 8.1 | 33.0–55.9 | 29.4, 59.0 |
|  | CMJ$_A$ | 53.4 ± 10.0 | 34.9–63.9 | 31.6, 67.0 |

Table 4. ANOVA Type I effects for the optimised models' outer validation RMSE.

| Effect | DF† | F | ω² |
|---|---|---|---|
| Model | 2 | 63.5 [46.4, 81.7] *** § | 0.106 [0.080, 0.134] |
| Jump Type | 1 | 55.1 [32.8, 84.9] *** § | 0.046 [0.028, 0.069] |
| Sensor | 3 | 7.7 [3.2, 14.0] *** § | 0.017 [0.006, 0.033] |
| Signal | 1 | 6.1 [0.9, 17.5] * | 0.004 [0.000, 0.014] |
| Model × Jump Type | 2 | 3.5 [0.4, 10.4] * | 0.004 [-0.001, 0.015] |

Significance for the central estimate indicated by $*$ $p < 0.05$, $**$ $p < 0.01$, $***$ $p < 0.001$.

§ indicates significance across the bootstrapped 90% CI shown in brackets.

† DF = Degrees of Freedom.

(This general comparison between model types will be revised as the models are refined below.) Considering the models based on the resultant signal for the $CMJ_{NA}$ (best combination for the jump type and signal representation conditions), the GPR models based on the LB sensor data yielded the lowest error (2.67 W·kg⁻¹, Fig 3E). As the LB-$CMJ_{NA}$ resultant data set yielded the best models, it was carried forward for the further optimisation of the three model types below.

The distribution of optimal parameters, aggregated over all outer folds, is shown in Fig 4 (data parameters) and Fig 5 (model parameters) for each model type. The ensemble optimal value for each parameter is highlighted on each plot (peak probability density or peak frequency). Most distributions are spread widely across the range with only a modest peak (e.g. $t_{pre}$ and $t_{post}$), but for some there are more prominent peaks (e.g. SVM model parameters; Fig 4B), none more so than the strong preference for no standardisation. Peaks in the optimisation parameter distributions reflect minima in the partial plots of the SM, as expected (Figs 5–7).

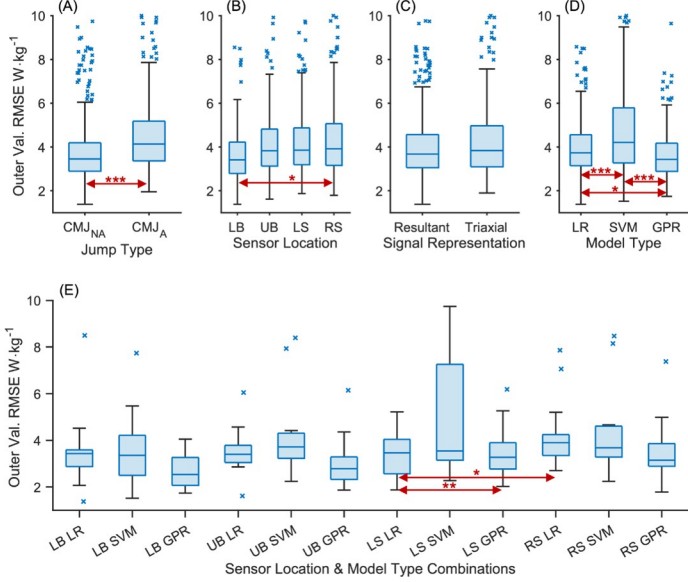

Fig 3. Outer validation RMSE distribution by level for each condition. Top row: single factors in the GLM, namely (A) Jump Type; (B) Sensor location; (C) Signal representation; (D) Model type. Bottom row: two factors (E) Model type and sensor location for the $CMJ_{NA}$ using the resultant signal representation. Horizontal arrows indicate significant differences, where $*$ $p < 0.05$, $**$ $p < 0.01$, $***$ $p < 0.001$.

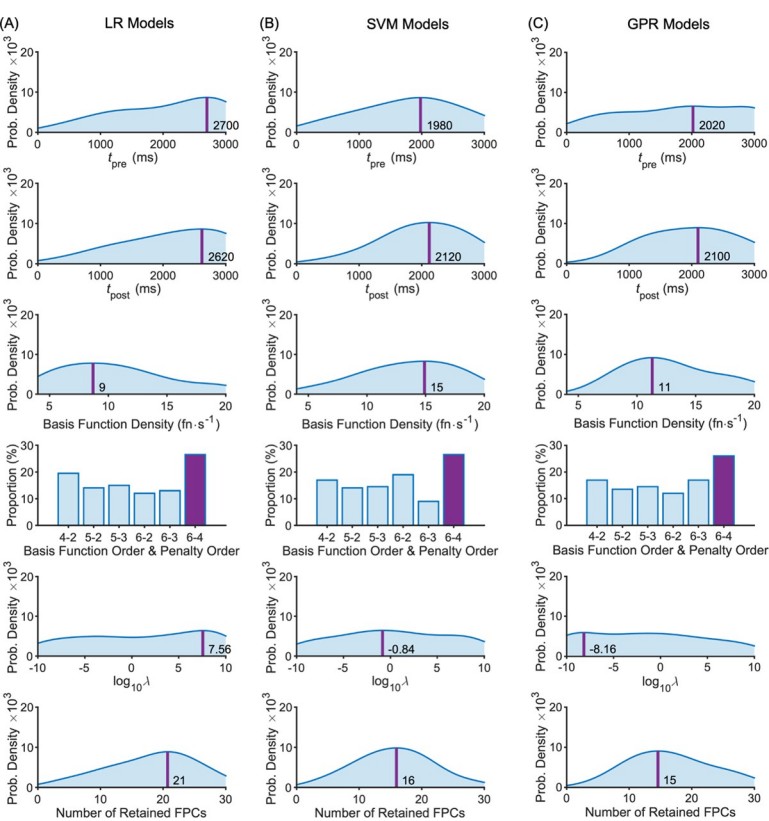

**Fig 4. Data parameter distributions across the intermediate models for the LB-CMJ$_{NA}$ data set with ensemble optimal values highlighted.** (A) LR model type; (B) SVM model type; (C) GPR model type. Optimal values are shown by the darker shaded bar for categorical parameters and by a darker vertical line at the peak position for numeric parameters with that optimal value shown.

The NCV predictive errors declined progressively with less variance between outer folds when the optimal parameters were refined (Table 5). The ranking between the three model types changed, resulting in the SVM model achieving the lowest predictive error of 2.27 W·kg$^{-1}$. In the final round there was no significant difference between the models' predictive error ($p > 0.860$). The LR model achieved marginally the lowest error, but all three were within 0.1 W·kg$^{-1}$ of one another (Table 6). In many cases, but not all, excluding specific parameters from the optimisation resulted in more peaked distributions, as can be seen in supplementary material, S1–S3 Figs.

## Discussion

This study developed ML models for estimating peak power in the CMJ from accelerometer data from a body-worn inertial sensor. We aimed to produce a model with a predictive error smaller than a typical athlete's inter-day variability. If that level of accuracy were achieved, such a field-based system could be used reliably for monitoring athletes' neuromuscular power. To this end, robust procedures were implemented to obtain unbiased estimates of how the models would perform on independent data. The best model achieved a generalised predictive error of 2.3 W·kg$^{-1}$ according to NCV and an independent error of 2.0 W·kg$^{-1}$ with the holdout data set. In percentage terms, these errors amount to 5.1% and 4.2% of the mean peak power. These errors are higher than the 3.4% target level for inter-day variability, determined *a priori* from three studies [10,60,61], as presented in the introduction. The 3.4% level is

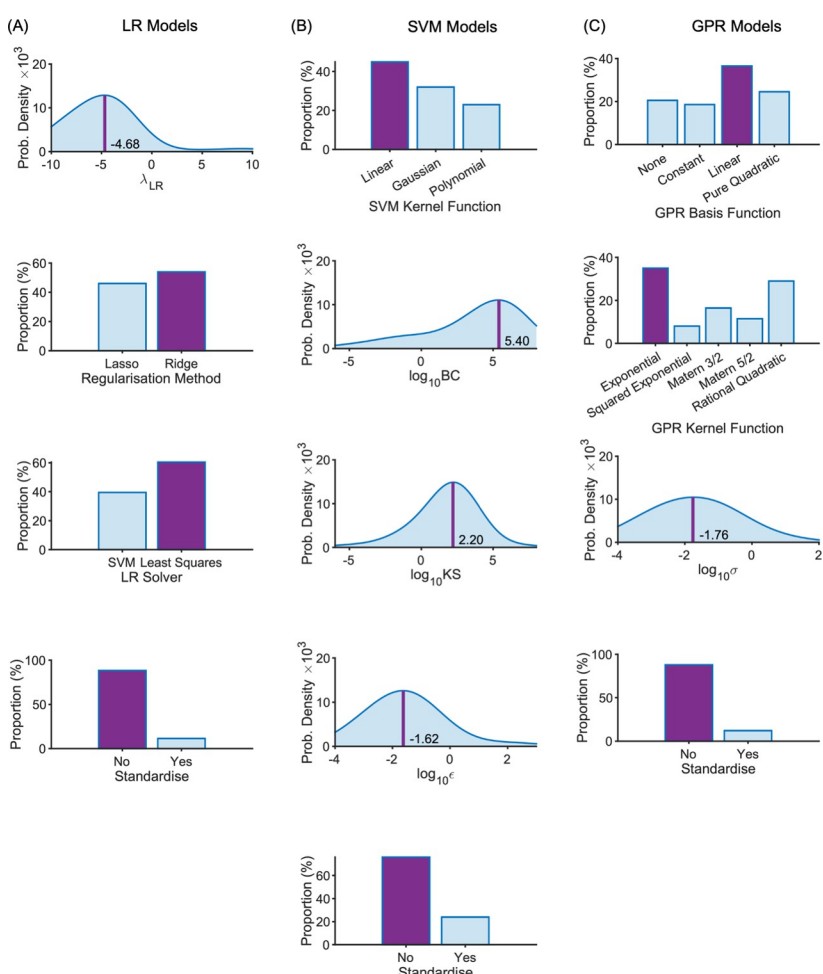

**Fig 5. Model parameter distributions across the intermediate models for the LB-CMJ$_{NA}$ data set with ensemble optimal values highlighted.** (A) LR model type; (B) SVM model type; (C) GPR model type. Optimal values are shown by the darker shaded bar for categorical parameters and by a darker vertical line at the peak position for numeric parameters with that optimal value shown.

equivalent to 1.55 W·kg$^{-1}$ with this data set. Thus, our sensor-based system and model does not meet the level of accuracy needed for practical day-to-day use.

Although our approach did not produce a sufficiently accurate model, the results are a considerable improvement over previous attempts in the literature. Estimates of peak power based on jump height had errors of 6.0–16.5% [20–25] while the Newtonian sensor-based calculations resulted in errors of 10.7–21.2% [28–30]. The lowest error reported in those studies was for the Canavan-Vescovi equation [21], but it was based on data from only 20 participants. In subsequent larger studies using the same equation, errors of 2.0%, 25.3% and 27.6% were reported [23,25,27]. The Sayers equation was the most consistent with errors of 5.3 ± 1.2 W·kg$^{-1}$ (10.5 ± 4.3%) across six studies [20–25]. These studies did not use similarly robust methods to estimate the expected error on independent data, as we did in our study, so their true generalised errors may in fact be higher.

It should also be noted that the performance levels achieved in our study are representative of those reported in the literature. For example, the CMJ$_{NA}$ mean power output of 48.4 W·kg$^{-1}$ for men in our study compares with 53.6 W·kg$^{-1}$ for professional rugby players [82], 54 W·kg$^{-1}$

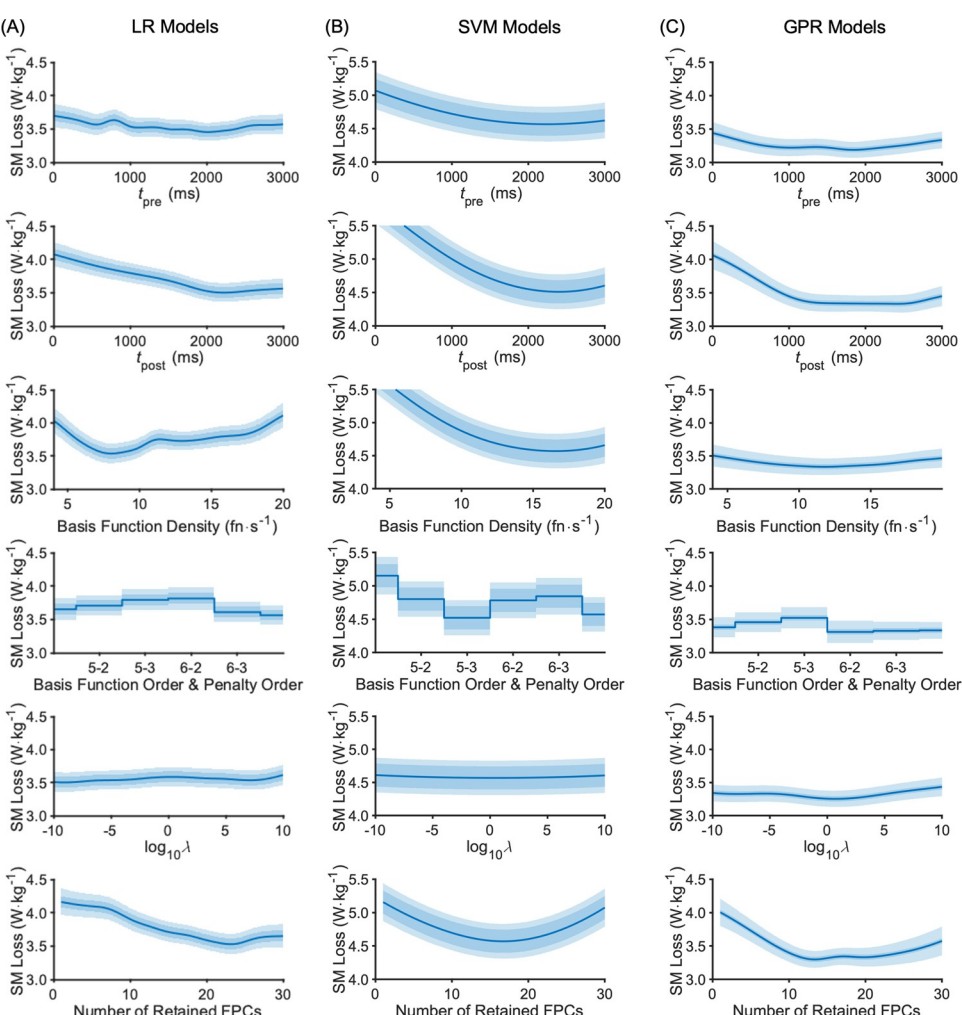

**Fig 6. Aggregated surrogate model partial plots for the data parameters from the LB-CMJ$_{NA}$ data set showing the predicted AM loss at the global minimum.** (A) LR model type; (B) SVM model type; (C) GPR model type. The central blue line is the central SM estimate. The darker shaded area about this line is the SM fitted noise level. The lighter shaded area covers the standard deviation. Note that for SVM (middle column), the SM range (y-axis) is higher.

for Australian rules football players [9], and 65.1 W·kg$^{-1}$ for college-level team-sport athletes [10]. Our female participants' mean performance of 38.2 W·kg$^{-1}$ places them in between the 34.8 W·kg$^{-1}$ reported for college students who played sports recreationally [83] and the 43.4 W·kg$^{-1}$ achieved by NCAA volleyball players [84].

## Conditions influencing the model

**Jump type.** The errors for the CMJ$_{NA}$ were significantly lower than for the CMJ$_A$ in absolute terms (0.8 W·kg$^{-1}$), but in relative terms they were much closer (0.5%) (Fig 3A). The additional degrees of freedom associated with arm swing makes peak power harder to predict, but only moderately so. It should be noted that these are comparisons between different (optimised) models on different data sets, not comparisons of how the same model performs on different data sets. This reveals the adaptability of model fitting and optimisation, as is reflected by jump type having a weak effect on the model and data processing parameters. Despite the arm swing introducing more degrees of freedom with the possibility of different swing

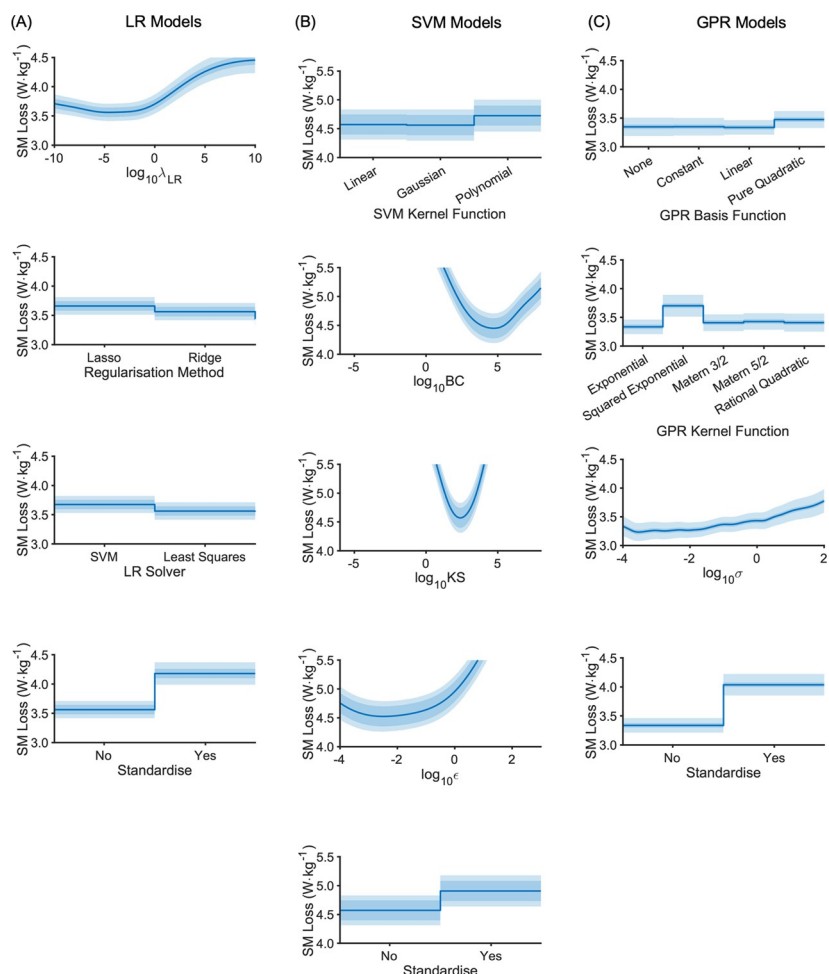

**Fig 7. Aggregated surrogate model partial plots for the model parameters from the LB-CMJ$_{NA}$ data set showing the predicted AM loss at the global minimum.** (A) LR model type; (B) SVM model type; (C) GPR model type. The central blue line is the central SM estimate. The darker shaded area about this line is the SM fitted noise level. The lighter shaded area covers the standard deviation. Note that for SVM (middle column), the SM range (y-axis) is higher.

movement patterns, the models could accommodate the greater complexity. This finding suggests that such a modelling approach may be suitable for estimating performance metrics in more complex movements.

**Table 5. Predictive error estimates over progressive optimisations for each model type using the resultant LB sensor for the CMJ$_{NA}$, based on nested cross validation and the independent holdout test.**

| RMSE (W·kg$^{-1}$) | LR | SVM | GPR |
|---|---|---|---|
| NCV– 1$^{st}$ round † | 3.50 ± 1.37 | 3.53 ± 1.44 | 2.67 ± 0.68 |
| NCV– 2$^{nd}$ round † | 3.11 ± 0.89 | 3.38 ± 2.11 | 2.59 ± 0.69 |
| NCV– 3$^{rd}$ round † | 2.93 ± 0.89 | 2.44 ± 0.47 | 2.47 ± 0.49 |
| NCV– 4$^{th}$ round † | 2.82 ± 0.87 | 2.27 ± 0.51 | 2.38 ± 0.54 |
| Holdout ‡ | 1.91 | 2.02 | 2.02 |

† For NCV (Nested Cross Validation) estimates, the mean loss is shown ± standard deviation over 20 outer folds. The standard errors in the final round estimates are 0.19 W·kg$^{-1}$, 0.12 W·kg$^{-1}$ and 0.11 W·kg$^{-1}$, respectively for LR, SVM and GPR.

‡The holdout test has a single error specific to that data set.

**Table 6. Ensemble optimal parameters over successive optimisations (1st, 2nd, 3rd, 4th) for each model type using the resultant LB sensor for the CMJ_NA.**

| Optimisation Round | LR | | | | SVM | | | | GPR | | | |
|---|---|---|---|---|---|---|---|---|---|---|---|---|
| | 1st | 2nd | 3rd | 4th | 1st | 2nd | 3rd | 4th | 1st | 2nd | 3rd | 4th |
| $t_{pre}$ (s) | 2.8 | 2.9 | 1.9 | 2.8 | 1.9 | 2.7 | 2.8 | 1.2 | 2.9 | 2.8 | 2.9 | 1.2 |
| $t_{post}$ (s) | 2.9 | 2.8 | 2.8 | 2.7 | 1.8 | 1.7 | 1.5 | 1.2 | 2.2 | 2.4 | 1.2 | 2.8 |
| $\rho$ (fn·s$^{-1}$) | 9 | 7 | ... | ... | 16 | 8 | ... | ... | 12 | 10 | 8 | ... |
| $F$ | 6–4 | 6–4 | 6–4 | ... | 6–4 | 6–4 | 6–4 | ... | 6–4 | 4–2 | 6–4 | ... |
| $\log_{10}\lambda$ | 8.8 | -9.0 | -9.5 | ... | -2.4 | -3.0 | 0.8 | -9.0 | -9.5 | -1.7 | -9.1 | -8.4 |
| $N_C$ | 21 | 20 | 24 | ... | 16 | 20 | 24 | ... | 16 | 12 | 23 | ... |
| $Z$ | No | ... | ... | ... | No | ... | ... | ... | No | ... | ... | ... |
| $R$ | Ridge | ... | ... | ... | | | | | | | | |
| $S$ | LSq | ... | ... | ... | | | | | | | | |
| $\log_{10}\lambda_{LR}$ | -4.0 | -1.6 | ... | -1.0 | | | | | | | | |
| $K_{SVM}$ | | | | | Linear | Gaussian | ... | ... | | | | |
| $\log_{10}$ BC | | | | | 5.8 | 3.1 | 3.3 | ... | | | | |
| $\log_{10}$ KS | | | | | 2.4 | 2.2 | ... | ... | | | | |
| $\log_{10}\varepsilon$ | | | | | -1.5 | -0.7 | ... | ... | | | | |
| $B$ | | | | | | | | | Linear | None | ... | ... |
| $K_{GPR}$ | | | | | | | | | Exp | Exp | ... | ... |
| $\log_{10}\sigma$ | | | | | | | | | -1.3 | -0.5 | ... | ... |

Successive optimisations progressively involve fewer parameters, narrowing the search and intensifying observations in promising regions. When a parameter's 'true' value has been determined, is it underlined and then held fixed in subsequent optimisations, indicated by the ellipsis. The 'true' optimal value is accepted when the parameter distribution shows a narrowly defined, unambiguous peak.

**Sensor location.** Placing a sensor on the lower back provided the most accurate estimates of peak power of the four anatomical locations considered. The LB models' mean errors were consistently lower than those based on sensor data from other locations. In biomechanics, the lower back tends to be used more often for sensor attachment, but it will depend on the application in question [66]. In the case of predicting peak power in vertical jumping, having a sensor close to the body's CM appears to be advantageous, as seen in our results, even though the CM does not have a fixed anatomical location. In comparison, the Newtonian approaches of previous investigations using inertial sensors [31–34] rely on the assumption that the sensor's movements match those of the body's CM. Even if those algorithms could perfectly correct for the sensor's changing orientation, the resulting peak power estimate would pertain to the motion of a sensor rather than the body as a whole. The sensor would have a fixed anatomical location while the body's CM would move dynamically relative to such a reference point. Hence, the differences in the trajectories of the body's CM and the sensor will be a source of error in Newtonian methods.

A machine learning approach, in contrast, compensates for different sensor positions in the fitting procedure, determining the best (linear) combination of features to approximate the outcome variable. Hence, models based on sensor data from other locations, seemingly less advantageous, were only slightly less accurate. Both the LB and UB sensors detect trunk movement, which makes the largest segmental contribution to the work done and take-off velocity in a vertical jump [85–87]. However, the same cannot be said for the shank sensors, although the LS/RS models were as accurate as the UB models. The LS/RS sensors tracked the shanks' changing inclination, a movement with fewer degrees of freedom. As with the comparison

between jump types, this is further evidence of the adaptability of a modelling procedure based on extracting patterns from the data.

In professional team sports, players often wear an inertial measurement unit on the upper back, which usually includes a GPS tracker. In principle, such a sensor could be re-purposed for peak power measurements, which would make it convenient for players and coaches as no additional setup would be required to attach a second sensor for a jump test. The UB model is less accurate than its LB equivalent, but the difference is only marginal. If further improvements could be made to feature extraction methods or the modelling procedure, then using such a sensor-based system for peak power measurement could become a realistic proposition provided the sensor is well-coupled to the player. Field-based testing of peak power could then be incorporated into training programmes, provided other limitations can be overcome, allowing many more tests to be conducted, which may in certain applications partly compensate for the lower level of accuracy compared to the force platform gold standard.

**Signal representation.** The models based on the resultant signal had marginally lower errors than their triaxial counterparts. The inertial accelerations would have been primarily vertical, making the resultant signal a reasonable first approximation. In principle, the triaxial models had more information, but with many more predictors the model was more prone to overfitting. Furthermore, the sensor's changing inclination in the sagittal plane would bias the accelerations measured along each axis. The baseline gravity vector would shift proportionally between the sensor's X- and Z-axes while the body's inertial acceleration moves in and out of alignment with those axes. However, orientation correction is not a requirement when using a pattern-based machine learning approach in our case, in contrast to the Newtonian approaches discussed above. Our models will have found the best weighting for the FPCs, thus implicitly compensating for the effects of changing sensor orientation, albeit imperfectly. Since the CMJ is a well-controlled movement, making it a valid and reliable test [18,88,89], the changing bias in the inertial accelerations would generally be consistent across jumps. However, differences in strength, coordination, fatigue, limb lengths and muscle morphology will account for variations in the movement pattern, limiting the accuracy of the models [90–93]. Whilst IMUs could correct for orientation, they have an inherent lag in responding to changes of orientation [31], which may limit their suitability for explosive movements. Further research would be needed to determine whether using IMUs rather than simple accelerometers could improve the model predictions, as the additional gyroscope data would permit a correction for sensor orientation [e.g. [94]].

**Model type.** The final condition was model type where common algorithms were considered, including parametric (LR) and non-parametric methods (SVM and GPR). After the first round of optimisation, the GPR appeared to be the best for this application, but further refinements revealed SVM achieved the lowest errors. This indicates that the global optimum for SVM was harder to find, as may be expected with non-parametric models, which are generally sensitive to the values of the kernel parameters. SVM had three strong parameters (BC, KS and $\varepsilon$), all with a continuous range. If one of those parameters was slightly adrift from the true optimum, the errors could be substantially higher. On the other hand, LR and GPR had only had one strong, real parameter each ($\lambda_{LR}$ and $\sigma$, respectively), but more categorical parameters that were easier to optimise. GPR was less prone to overfitting with its Bayesian approach, choosing the most likely solution from the distribution of possible fits. In contrast, SVM had a propensity to produce wildly inaccurate predictions if its hyperparameters were chosen poorly, hence the need to Winsorise the estimates. Furthermore, SVM fitting could occasionally be time-consuming due to its kernel-type design, as indicated by AM execution times: median 0.187 s, 90% CI [0.095 s, 5.080 s]. The times for LR and GPR were more consistently shorter overall: 0.133 s, [0.079, 0.221] s and 0.185 s, [0.122, 0.283] s for GPR. In summary, SVM optimisation

was time-consuming and at times unreliable, but it produced the best estimates in the end. GPR models were more forgiving, less prone to overfitting and easier to work with in practical terms. Ultimately, if the exploration of parameter space is thorough and properly directed, which was achieved by narrowing the parameter search ranges, then the challenges of optimisation with these models can be met.

## Optimal parameter values

The optimal parameter values provide answers for our third research question on the best data processing setup. In optimising the time window, the model needs to define a period that includes all the relevant information for the prediction, but there is a trade-off. Extending the time window provides more information, but it comes with the risk of overfitting. A longer period will increasingly encompass periods outside of the jumping movement, especially for jumps that are performed more quickly than others. In these periods, the only inertial accelerations should be due to body sway in standing. In all cases, the optimal time window extended beyond take-off to include flight and landing, indicating valuable information in this latter portion of the signal relating to mechanical power generation and dissipation. The final SVM model may have outperformed the others because its window [-1.2 s, 1.2 s] was limited to these more substantial inertial accelerations, making it less prone to overfitting.

The flight time itself may be useful as the first FPC, which had the highest correlation with peak power, mainly described variations in the timing of the landing impact spike ($580 \pm 65$ ms after take-off compared to an actual flight time according to the VGRF data of $480 \pm 66$ ms). The second FPC primarily described variations in the impact spike amplitude, indicating an association with peak power via jump height. The models in our study made more accurate predictions than the peak power formulae from previous research because using several FPCs as predictors provides more information than flight time alone. To verify this, we fitted a simple regression model based on flight time and body mass, the same as those previous peak power formulae, and obtained a cross-validated RMSE of 3.49 W·kg$^{-1}$.

The roughness penalty, $\lambda$, controls how much the signal is smoothed, but its final optimal value ($< 10^{-8}$) was very low (Table 6). In comparison, generalised cross validation, the standard method of determining the roughness penalty, yielded $10^2$ [42]. Light smoothing preserves the amplitude of sharp peaks, particularly the impact acceleration spike on landing. It appears the modelling procedure relied partly on reducing the basis function density, $\rho$, to control complexity. It was helped by using 6$^{th}$ order b-splines, which made up for the low density with considerable flexibility, not just from the quintic polynomials but from their high degree of overlap. The low densities reduced the FPCA computational cost considerably, which is roughly proportional to the square of the number of basis functions. In summary, functional smoothing had quite a limited role in controlling complexity. Indeed, it was optimal to retain a large number of components from FPCA, many of which described very small signal variations.

Having a long list of potentially complex features appeared to be tolerable because in part the models had their own ways of regulating complexity. The LR model favoured ridge regression, which reduced coefficients through the regularisation parameter $\lambda_{LR}$, diminishing the influence of some features. The final SVM model had a narrow support vector margin ($\varepsilon$), facilitated by a high box constraint (BC) or soft margin, making the model more flexible and less prone to overfitting. At one level, the GPR model used the fitted noise level ($\sigma$) of $10^{-0.5}$ (~ 0.3 W·kg$^{-1}$), but overfitting was controlled mainly through its Bayesian approach. The other part of the explanation can be attributed to the unrotated FPCs having an inherent reduction in amplitude with each successive component. Finally, the optimiser favoured no feature

standardisation because the influence of higher-order FPCs diminished, thus providing a natural form of regularisation.

## Modelling procedures

Cross validation is widely regarded as an essential element of machine learning, yet there are comparatively few examples of nested cross validation in the literature. In our study, twice-repeated 10-fold CV (20 outer folds) produced reasonable estimates of the expected generalised error with a standard error of ~ 0.15 W·kg$^{-1}$. More iterations could refine this estimate, but it is already small enough to make no meaningful difference in practice. However, the expected value should not obscure the fact that there was considerable variation in error between folds, indicating a high degree of model sensitivity to the data. It follows that the error for any given jump is somewhat uncertain. Only in aggregate with the large samples can model performance be assessed with the precision reported above.

The statistical model comparing different conditions could only account for 19% of the outer validation errors. The unexplained variance can be attributed to the subsampling variation of the CV inner loop and to differences in the distributions between the inner training set and the outer validation set. The variance could be reduced by averaging over more CV repetitions [62,57,95,96], but that would come with a higher computational cost. For example, the AM loss with two-fold CV without repetition had a noise level of 0.577 W·kg$^{-1}$, while five repeats reduced noise to 0.258 W·kg$^{-1}$, but the execution time rose by a factor of 4.1.

Optimising a noisy objective function would typically be the task of a Bayesian optimiser. However, although the search directed by its expected-improvement algorithm (or similar) is highly efficient, it comes with a high overhead that rises steeply as more observations are added, as others have reported [97]. We found MATLAB's *bayesopt* optimiser exceeded the AM cost by a factor of 10 after just 50 iterations. Researchers have previously investigated more efficient Bayesian alternatives, but the overhead remains significant [98–100]. The overhead with our method, including SM fitting and PSO, was only 3.5% of the total execution time, allowing a high proportion of computing resources to be devoted to the search.

## Limitations

The models depended on accelerometer signals being aligned perfectly with take-off, which had been achieved by referring to the synchronised VGRF data. If an accelerometer-based system were to be implemented, it would have to be self-sufficient by detecting take-off from the accelerometer data alone. That would introduce an alignment error, which could potentially reduce the effectiveness of FPCA, depending on the algorithm's accuracy [42]. Algorithms for estimating CMJ flight time from body-worn inertial sensors have errors of 21–37 ms [34,101,102]. Assuming the take-off and landing detection errors have identical normal distributions, the take-off errors would be 15–26 ms. Further research is needed to develop a suitable algorithm and quantify its effect on the AM validation error.

FPCA as a feature extraction method is based on a linear decomposition that requires more components to represent a pattern than would otherwise be the case with nonlinear representations, such as those obtained using autoencoders [103]. Using such feature encodings may improve the models, although it may be more appropriate to use a second neural network to make the performance predictions. Such an approach may work well in more complex situations where athletic movements have more degrees of freedom. What our study has shown is that reasonably accurate estimates can be obtained using linear feature representations provided the movement is carefully controlled in a test environment.

Finally, the NCV error estimates assumed independent, identically distributed data, an assumption that is common in machine learning. Were the model applied to a new cohort with a different peak power distribution to the one used here, the errors would have a different spread making the RMSE perhaps higher or lower. This can be seen in the holdout errors where the LR model outperformed the other two and its NCV estimate. Recruiting participants from a range of sports partly addressed this as it created a heterogeneous data set without being specific to a single cohort. A replication study evaluating the same methodology with different sensors, researchers and participants would contribute greatly to the ecological validity of the research.

## Conclusions

The final models developed in this study using accelerometer data from body-worn sensors predicted peak power in the $CMJ_{NA}$ more accurately than has hitherto been achieved by a field-based system. The error estimates reported above can be considered realistic owing to the robust procedures implemented. However, with errors of 2.3 W·kg$^{-1}$ or 5.1%, they do not reach the level of accuracy desired for practical use. Nevertheless, with further developments, this gap may be bridged such that a valuable single-sensor system could be applied for certain practical applications. The models themselves were based on FPCA, which has been successful in biomechanics, with optimisation of data processing parameters, as well as the model's hyperparameters. We believe this is the first biomechanics study to take this comprehensive approach to optimisation. In yielding a small number of features characterising time series data, FPCA allows classical machine learning models to be employed. It would be suitable where there is a natural point of alignment, such as jump take-off, so the modes of variation become apparent without further data manipulation. It is a modelling approach that has potentially wider applications in biomechanics as it has been shown to be adaptable to different data sets.

## Supporting information

**S1 Fig. LR model optimal parameter distributions over four successive optimisations.** Parameters may be eliminated in successive rounds if there is a clear preference for an optimal value. Alternatively, the range of possible values may be reduced. In doing so, subsequent distributions tend to have more prominent peaks, but not always, as with the time window parameters. Abbreviations. Vertical axes: Proportion = Proportion (%); Density = Probability Density Function × 10$^3$. Standardise Axis: N = No; Y = Yes. Regularisation Axis: L = Lasso; R = Ridge. LR Solver Axis: S = SVM; L = Least Squares.
(TIF)

**S2 Fig. SVM model optimal parameter distributions over four successive optimisations.** Parameters may be eliminated in successive rounds if there is a clear preference for an optimal value. Alternatively, the range of possible values may be reduced. Abbreviations. Vertical axes: Proportion = Proportion (%); Density = Probability Density Function × 10$^3$. Standardise Axis: N = No; Y = Yes. SVM Kernel Axis: L = Linear; G = Gaussian; P = Polynomial.
(TIF)

**S3 Fig. GPR model optimal parameter distributions over four successive optimisations.** Parameters may be eliminated in successive rounds if there is a clear preference for an optimal value. Alternatively, the range of possible values may be reduced. Abbreviations. Vertical axes: Proportion = Proportion (%); Density = Probability Density Function × 10$^3$. GPR Basis Axis: N = None; C = Constant; L = Linear; RQ = Rational Quadratic. GPR Kernel Axis:

E = Exponential; SE = Squared Exponential; M3 = Matérn 3/2; M5 = Matérn 5/2;
RQ = Rational Quadratic. Standardise Axis: N = No; Y = Yes.
(TIF)

**S4 Fig.**
(TIF)

## Author Contributions

**Conceptualization:** Mark G. E. White, Neil E. Bezodis, Huw Summers.

**Data curation:** Mark G. E. White.

**Formal analysis:** Mark G. E. White.

**Funding acquisition:** Huw Summers.

**Investigation:** Mark G. E. White.

**Methodology:** Mark G. E. White, Jonathon Neville, Paul Rees.

**Project administration:** Neil E. Bezodis.

**Software:** Mark G. E. White.

**Supervision:** Neil E. Bezodis, Jonathon Neville, Paul Rees.

**Validation:** Mark G. E. White.

**Writing – original draft:** Mark G. E. White.

**Writing – review & editing:** Neil E. Bezodis, Jonathon Neville, Paul Rees.

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
