## [Decision Letter · Decision Letter 0]

25 Nov 2021

PONE-D-21-30596Determining jumping performance from a single body-worn accelerometer using machine learningPLOS ONE

Dear Dr. White,

Thank you for submitting your manuscript to PLOS ONE. After careful consideration, we feel that it has merit but does not fully meet PLOS ONE’s publication criteria as it currently stands. Therefore, we invite you to submit a revised version of the manuscript that addresses the points raised during the review process.

We look forward to receiving your revised manuscript.

Kind regards,

Chris Connaboy

Academic Editor

PLOS ONE

3. 1.  Author - How to Upload a Striking Image in EM

You can choose to upload a striking image in Editorial Manager when you submit your manuscript. The image must be derived from a figure or supporting information file from your submission. To upload a striking image use the drop down menu on the “Attach Files” page to select “Striking Image” then select the image you would like to represent your manuscript. The striking image will not appear in the PDF sent to reviewers and editors, so it is important to make sure all necessary figures for the review process are uploaded as separate "Figure" file types.

Once your manuscript is accepted for publication, this image file will represent your article on the PLOS ONE homepage

3.2.  Author - What can be Uploaded as a Striking Image

Your striking image file will represent your article upon publication on the PLOS ONE homepage. The image must be derived from a figure or supporting information file from your manuscript. Ideally, striking images should be high resolution, eye-catching, single panel images that do no contain additional text, scale bars, or arrows.

Please also keep in mind that PLOS's Creative Commons Attribution License applies to striking images. As such, please do not submit any figures or photos that have been previously copyrighted unless you have express written permission from the copyright holder to publish under the CCAL license. You can read more about PLOS’s Creative Commons License on our homepage: http://journals.plos.org/plosone/s/licenses-and-copyright .

Reviewers' comments:

Reviewer's Responses to Questions

**Comments to the Author**

1. Is the manuscript technically sound, and do the data support the conclusions?

Reviewer #1: Yes

Reviewer #2: Yes

2. Has the statistical analysis been performed appropriately and rigorously? 

Reviewer #1: Yes

Reviewer #2: Yes

3. Have the authors made all data underlying the findings in their manuscript fully available?

Reviewer #1: Yes

Reviewer #2: Yes

4. Is the manuscript presented in an intelligible fashion and written in standard English?

Reviewer #1: Yes

Reviewer #2: Yes

5. Review Comments to the Author

Reviewer #1: The authors investigated the accuracy of different machine learning models to predict peak power output during vertical jumps using acceleration data derived from inertial sensors placed at various anatomical positions. The research questions that the authors were answering are appropriate given the current interest in practical measurement devices within the field of strength and conditioning. Furthermore, the outcomes of the authors' study present some clear practical recommendations for both researchers and practitioners alike. Overall, the manuscript is well-written and the findings are clearly presented. However, I do have some minor issues that I would like the authors to address.

General comment

There are many abbreviations used by the authors throughout the manuscript, many of which the reader may not be entirely familiar with. As such, the authors may consider presenting a list of abbreviations at the beginning of their paper to assist the reader.

Specific comments

Line 48: Make it clear that you are proposing that coaches/practitioners prefer to use jump height instead of peak power output.

Line 85: You present the abbreviation CV here without defining it (presumably cross-validation).

Line 144: VGRF has already been defined in line 66.

Line 135: You note the method used to identify take-off. However, in line 538 you discuss flight time (although this data is not presented in the Results) and so do you need to also present your method of determining landing here?

Line 180: CV has already been defined in line 85.

Table 4: Explain what these values are (presumably means +/- SD).

Line 433: Consider changing to "Estimates of peak power based on jump height..."

Line 436: Consider changing to "In subsequent larger studies using the same equation, errors of 2.0%, 25.3% and 27.6% were reported..."

Line 438: Provide the reference numbers for these 6 studies.

Line 461: Consider changing to "Placing a sensor on the lower back provided the most accurate peak power of the four anatomical locations considered..."

Line 588: Change to "Previous researchers have investigated more efficient Bayesian alternatives..."

Reviewer #2: The manuscript investigates the use of a machine learning approach for improving the peak power estimates obtained via accelerometer measures with different countermovement jump paradigms and sensor locations.

The methodological part, comprising modeling and statistical aspects, is punctually described, and the results are clearly listed. Appreciation should be addressed to the use of the nested cross-validation approach, enforcing further model generalization outside the presented dataset.

The main concern regards the biomechanical description of the investigated motor task. Inaccuracies emerge when describing how the power was computed. Being the only variable to be investigated, such a description requires expansion for both reader and study clarity. Moreover, some assumptions related to the center of mass seem inappropriate and deserve further emphasis in the limitation's discussion. Nonetheless, study limitations are well known to the authors, and they are clearly explained in the discussion section. Moreover, it is highlighted in the text the fact that this experimental setup is part of a wider project, maybe constraining subject testing. This point may however become explicit, if true.

In the following sections, the main comments are listed, sorted on the basis of their relevance.

Major Compulsory Revisions

Introduction

Line 90-93: It seems to me not correct to report this in the introductory section. It would be beneficial to carry out this consideration later in the discussion section, where you can make “numerical” comparisons with similar studies found in literature.

Materials and methods

Line 114-116: What Owen and colleagues did to compute the instantaneous power was not a double integration. They body-weight-normalized the VGRF, hence obtaining the vertical acceleration, equaling zero with the subject standing still prior the jump. Starting from it, they integrated it once in order to obtain the vertical velocity. Finally, the product of velocity and VGRF was used to obtain the instantaneous power. However, the power they computed was not normalized to body mass. Normalization was necessary as the numerical integration of the acceleration was the only way to compute velocity. Please, rearrange this part accordingly.

Discussion

Line 423-426: This seems in contrast with the goal of this study declared in the introductory section (Line 90-93 reported above).

Line 463-466: Even though the CM has not a fixed anatomical location, one should consider that, when computing power through FP, one is actually computing the CM kinematics, irrespective of whether it is located on the FP. This is true as long as the jump is performed onto the FP sensing area, which is a pre-requisite of a proper jump analysis. Moreover, besides the performance parameters one can extract from different sensors, that specific location (L3-L5 vertebrae) allows for comparisons between instrumentations (e.g. FP Vs. IMU). Having said that, this sentence should be rephrased accordingly.

Line 498-499: This is not correct, since you must always correct for trunk bending when using IMUs. (https://doi.org/10.1111/sms.13546 - Eq. 1).

Minor Essential Revisions

Introduction

Line 47-49: This seems to be a strong assertion. Could you expand on this? To this aim, maybe a more general reference is required. The one you used refers to rugby players only.

Line 60-62: What you are saying is very true. Notwithstanding, a reference at the end of the sentence would be beneficial. I can suggest you this one: http://dx.doi.org/10.1080/02640414.2010.523089

Materials and Methods

Line 117-118: Can you provide the reader more information about the sensors you used? At least sampling frequency and full-scale range would be required.

Line 131-132: Not clear. May you expand a bit on this?

Line 134-135: What was done for the accelerometer signals? Did you choose the same take-off instant as the FP?

Line 155: Table 1 is poorly rendered. Did you considered to use the landscape layout to insert it?

Line 267-268: Why did you not simply exclude the outliers? The Winsorization process, in my opinion, tends to modify the data as it replaces arbitrarily the outliers with samples at fixed value. This is to be avoided if the method is to be used in unsupervised contexts, even though the number of outliers is very small compared to the whole dataset.

Results

Line 385-386: Table 5 bad rendering. See suggestion for Table 1.

Discussion

Line 467-469: I am not sure of what you are asserting here. Which is the reason why you are relating sensor orientation to the external mechanical power computation?

Discretionary Revisions and typos

Material and Methods

Line 121-122: Maybe a picture of the setup would be beneficial to show the sensor attachment technique. I am saying that as it would be useful for the experimental setup repeatability.

Results

Line 387-391: Maybe this paragraph has a more "methods" fashion.

6. PLOS authors have the option to publish the peer review history of their article (what does this mean?). If published, this will include your full peer review and any attached files.

Reviewer #1: No

Reviewer #2: No

---

## [Author Response · Author response to Decision Letter 0]

15 Dec 2021

Editor:

• Authors' Response: We have addressed the five areas where we believe our manuscript was not compliant with your requirements: (1) reformatting of the authors’ affiliations, including their contributions, which had been omitted; (2) naming the ethics committee fully, but otherwise our original ethics statement noted participants had given their informed, written consent; (3) including page numbers; (4) changing the name of manuscript file, but otherwise we believe the filenames comply with your requirements; (5) replacing the striking image with a new one without text. If, despite our diligence, some other aspect of the manuscript still remains non-compliant, we would be grateful if you would draw our attention to it.

Reviewer #1:

The authors investigated the accuracy of different machine learning models to predict peak power output during vertical jumps using acceleration data derived from inertial sensors placed at various anatomical positions. The research questions that the authors were answering are appropriate given the current interest in practical measurement devices within the field of strength and conditioning. Furthermore, the outcomes of the authors' study present some clear practical recommendations for both researchers and practitioners alike. Overall, the manuscript is well-written and the findings are clearly presented. However, I do have some minor issues that I would like the authors to address.

• Authors’ Response: Thank you for taking the time to review our papers and for your positive comments. We are pleased that you find our manuscript well-written and clearly presented, and that the practical recommendations are clear for both researchers and practitioners. We will address each of your comments in turn below and have made amendments to the manuscript using the Track Changes feature. All line numbers included in our responses below correspond to those on the new tracked changes version of the revised manuscript, as rendered in the PDF Submission. 

• Note that these line numbers differ from the Word version of the revised manuscript due to differences in page layout. However, the largest variance is no more than 24 lines by the conclusion section – the Word document has higher line numbers than the PDF submission document. The quoted line numbers will also differ to a greater extent from those in the clean version of the manuscript, which does not show the changes.

General comment

There are many abbreviations used by the authors throughout the manuscript, many of which the reader may not be entirely familiar with. As such, the authors may consider presenting a list of abbreviations at the beginning of their paper to assist the reader.

• Authors’ Response: A table of abbreviations has been included at the end of the introduction. See lines 109-110 of the revised manuscript in the PDF Submission.

Specific comments

Line 48: Make it clear that you are proposing that coaches/practitioners prefer to use jump height instead of peak power output.

• Authors’ Response: We have clarified this point with the words, “Many coaches prefer to measure jump height instead of peak power...”. (Please see line 53)

Line 85: You present the abbreviation CV here without defining it (presumably cross-validation).

• Authors’ Response: Yes, this is correct. A definition has been added here (line 91), and this acronym is included in the abbreviations table.

Line 144: VGRF has already been defined in line 66.

• Authors’ Response: We have removed the second definition from line 126 of the revised manuscript.

Line 135: You note the method used to identify take-off. However, in line 538 you discuss flight time (although this data is not presented in the Results) and so do you need to also present your method of determining landing here?

• Authors’ Response: Yes, thank you for this good suggestion since flight time enters the discussion. We have included a definition and provided flight time summary statistics in the discussion where it is addressed (lines 175-177). Including those times in the results section felt out of place, where at that point the focus is squarely on the peak power.

Line 180: CV has already been defined in line 85.

• Authors’ Response: “2-fold cross validation (CV)” has been amended to “2-fold CV” on line 222.

Table 4: Explain what these values are (presumably means +/- SD).

• Authors’ Response: Yes, that is correct for the NCV estimates. We have added the symbol † to all of the relevant row headings on line 423 to draw the reader’s attention to the table subtext, which confirms this (now Table 5).

Line 433: Consider changing to “Estimates of peak power based on jump height...”

• Authors’ Response: We have changed this on line 475.

Line 436: Consider changing to “In subsequent larger studies using the same equation, errors of 2.0%, 25.3% and 27.6% were reported...”

• Authors’ Response: Yes, this is better wording, thank you. We have updated the text (now on lines 478-479).

Line 438: Provide the reference numbers for these 6 studies.

• Authors’ Response: Added on line 480.

Line 461: Consider changing to “Placing a sensor on the lower back provided the most accurate peak power of the four anatomical locations considered...”

• Authors’ Response: Agreed, that is a clearer statement – thank you. We have modified the new wording slightly in lines 502-503: “Placing a sensor on the lower back provided the most accurate estimates of peak power of the four anatomical locations considered.”

Line 588: Change to “Previous researchers have investigated more efficient Bayesian alternatives...”

• Authors’ Response: Agreed, that is more appropriate. We have amended the word order slightly so “previously” precedes “investigated” on lines 643-644.

 

Reviewer #2:

The manuscript investigates the use of a machine learning approach for improving the peak power estimates obtained via accelerometer measures with different countermovement jump paradigms and sensor locations.

The methodological part, comprising modeling and statistical aspects, is punctually described, and the results are clearly listed. Appreciation should be addressed to the use of the nested cross-validation approach, enforcing further model generalization outside the presented dataset. The main concern regards the biomechanical description of the investigated motor task. Inaccuracies emerge when describing how the power was computed. Being the only variable to be investigated, such a description requires expansion for both reader and study clarity. Moreover, some assumptions related to the center of mass seem inappropriate and deserve further emphasis in the limitation’s discussion. Nonetheless, study limitations are well known to the authors, and they are clearly explained in the discussion section. Moreover, it is highlighted in the text the fact that this experimental setup is part of a wider project, maybe constraining subject testing. This point may however become explicit, if true.

• Authors’ Response: Thank you for taking the time to review our manuscript and for your positive comments. We will address each of your comments in turn below and have made amendments to the manuscript using the Track Changes feature. All line numbers included in our responses below correspond to those on the new tracked changes version of the revised manuscript, as rendered in the PDF Submission. 

• Note that these line numbers differ from the Word version of the revised manuscript due to differences in page layout. However, the largest variance is no more than 24 lines by the conclusion section – the Word document has higher line numbers than the PDF submission document. The quoted line numbers will also differ to a greater extent from those in the clean version of the manuscript, which does not show the changes.

Major Compulsory Revisions

Introduction

Line 90-93: It seems to me not correct to report this in the introductory section. It would be beneficial to carry out this consideration later in the discussion section, where you can make “numerical” comparisons with similar studies found in literature.

• Authors’ Response: Thank you for this suggestion. We considered at length where this would be best placed; we felt it was important to determine an appropriate error-level for a sensor-based system to have practical value a priori and thus felt it was important to include this here to best reflect our approach. This allowed us to assess the initial results, which fell short of this threshold value (3.4%), prompting further rounds of optimisation, as reported in what is Table 5 of the revised manuscript. This approach has similarities to determining the sample size for hypothesis testing, which involves estimating effect size, a priori. Thus, we are confident that it was appropriate to have determined in advance what level of accuracy would be required as it would not bias the model predictions but encourage an approach of continual improvement. We have amended the wording slightly, over lines 99-101 in the revised manuscript to more clearly reflect and explain our approach to the reader.

Materials and methods

Line 114-116: What Owen and colleagues did to compute the instantaneous power was not a double integration. They body-weight-normalized the VGRF, hence obtaining the vertical acceleration, equaling zero with the subject standing still prior the jump. Starting from it, they integrated it once in order to obtain the vertical velocity. Finally, the product of velocity and VGRF was used to obtain the instantaneous power. However, the power they computed was not normalized to body mass. Normalization was necessary as the numerical integration of the acceleration was the only way to compute velocity. Please, rearrange this part accordingly.

• Authors’ Response: Yes, you are quite correct. We did carry out the calculations as described, which differed from those described by Owen et al., as you rightly pointed out. We had cited the paper by Owen et al. because we had used a modified version of his technique for determining the point of jump initiation. Therefore, we have added more detail on our calculations (lines 127-132) whilst making clear that our reference to Owen et al. relates to their method of determining jump initiation (lines 132-136).

Discussion

Line 423-426: This seems in contrast with the goal of this study declared in the introductory section (Line 90-93 reported above).

• Authors’ Response: Yes, the stated aim of our study was not achieved in terms of developing a model which produced a predictive error smaller than the typical inter-day variability determined a priori. This approach (optimisation of FPCA models based on accelerometer data), as demonstrated, could not yield peak power predictions with a sufficiently low error that could be of practical use. However, given our novel and robust methods, and the fact that the accuracy achieved was an improvement on previous attempts, which had applied different theoretical approaches, we are confident that this is a valuable contribution to the understanding in this area. We have amended the wording to include the reference to this a priori target error level (lines 465-469) and have clarified the aim in the discussion opening (line 457-461). 

Line 463-466: Even though the CM has not a fixed anatomical location, one should consider that, when computing power through FP, one is actually computing the CM kinematics, irrespective of whether it is located on the FP. This is true as long as the jump is performed onto the FP sensing area, which is a pre-requisite of a proper jump analysis. Moreover, besides the performance parameters one can extract from different sensors, that specific location (L3-L5 vertebrae) allows for comparisons between instrumentations (e.g. FP Vs. IMU). Having said that, this sentence should be rephrased accordingly.

• Authors’ Response: We agree that when working with FP data, the displacement of the body’s CM and hence its kinematics can be obtained. However, we were unclear in our reference to Newtonian approaches, which was meant to relate to methods employed by previous studies where power or jump height was determined from sensor kinematic estimates. We believe that has led to your comment here. The sensor location at L3-L5 provides the best approximation for the CM, but the further point we wished to make was that even if the sensor’s kinematics were error-free, the performance metric (jump height or peak power) would pertain to the sensor rather than the body. As you correctly suggest, no single body-worn sensor can directly experience the kinematics of the body’s CM because the sensor is held in a fixed position on the body whereas the CM location is dynamic with reference to any fixed anatomical location. Those sensor-based Newtonian approaches are hampered by this fundamental limitation and therefore cannot provide very accurate estimates of jump performance. Hence, in general terms a model is needed to compensate for this limitation. We have revised our discussion of these points accordingly (lines 509-516).

Line 498-499: This is not correct, since you must always correct for trunk bending when using IMUs. (https://doi.org/10.1111/sms.13546 - Eq. 1).

• Authors’ Response: We used a pattern-based approach from machine learning where orientation is not necessarily critical, although we acknowledge it may help to some extent. The resultant models, which eschew orientation entirely, were the most accurate in our investigation. Still, perhaps the triaxial models might have performed better had an orientation correction algorithm had been employed. We cannot know for sure since our sensors were simple accelerometers without a gyroscope that is required for Madgwick’s algorithm. Our models will have found the correct weighting for the FPCs, implicitly compensating for the changing orientation, albeit imperfectly. What we demonstrated was that with this approach it is possible to estimate jump performance without necessarily correcting for orientation. It is an open question whether and to what extent our peak power estimates would be more accurate when using an IMU that would permit orientation correction. We have now acknowledged this more clearly on lines 554-557. 

• In the cited paper, Rantalainan et al. found that flight time, as detected from the sensor data, was preferable to estimating vertical take-off velocity. The latter method may be considered a Newtonian approach as it relies on maintaining an accurate determination of sensor orientation. However, Rantalainan et al. found it more error-prone than the flight time method, as others discovered previously [1–3]. This suggests that a full-IMU approach may not necessarily yield a meaningful improvement in accuracy that might be expected. Such algorithms are slow to respond to rapid orientation changes [4–6], as may occur in CMJ. Indeed, these considerations prompted us originally to consider a machine learning approach.

• We have amended the discussion to address these points above in response to your comment. Please see lines 544-553.

• The references (i.e. 1-6 cited above, which are not the same as 1-6 in the manuscript) can be found at the end of this document.

Minor Essential Revisions

Introduction

Line 47-49: This seems to be a strong assertion. Could you expand on this? To this aim, maybe a more general reference is required. The one you used refers to rugby players only.

• Authors’ Response: Thank you for pointing that out. We have cited another study instead reporting on a survey of 14 sports in Australia where all respondents who provided detailed information (11) stated that jump height was their chosen performance metric for the CMJ (reference [14] on line 54). Note that a change of citation does not show up tracked changes when using our referencing software. 

Line 60-62: What you are saying is very true. Notwithstanding, a reference at the end of the sentence would be beneficial. I can suggest you this one: http://dx.doi.org/10.1080/02640414.2010.523089

• Authors’ Response: Thank you. A reference to Picerno et al. has been added at the end of the paragraph on line 68, reference [34]. Note that a change of citation does not show up tracked changes when using our referencing software.

Materials and Methods

Line 117-118: Can you provide the reader more information about the sensors you used? At least sampling frequency and full-scale range would be required.

• Authors’ Response: We added more details about the range and the analogue transmission (line 142), sampled at 250 Hz. The sentences have been re-ordered, so mention of the 250 Hz sampling rate immediately follows the description of the analogue transmission (lines 142-144, 149-151). We also explain that the analogue signal made synchronisation possible.

 

Line 131-132: Not clear. May you expand a bit on this?

• Authors’ Response: We have expanded this description of the padding method to make it clearer (see lines 167-174).

Line 134-135: What was done for the accelerometer signals? Did you choose the same take-off instant as the FP?

• Authors’ Response: Yes, we used the same take-off instant from the VGRF data. The description has been updated accordingly to make this clear on lines 175-176.

Line 155: Table 1 is poorly rendered. Did you considered to use the landscape layout to insert it?

• Authors’ Response: We were attempting to comply with PLOS ONE’s guidelines, but on review we realise we have more scope for table formatting. Accordingly, we have reduced the size of the fonts and used landscape for the wide tables. Table 2 of the revised manuscript at line 197 has been reformatted accordingly.

Line 267-268: Why did you not simply exclude the outliers? The Winsorization process, in my our opinion, tends to modify the data as it replaces arbitrarily the outliers with samples at fixed value. This is to be avoided if the method is to be used in unsupervised contexts, even though the number of outliers is very small compared to the whole dataset.

• Authors’ Response: Thank you for this important observation. We did not believe that we had a good justification for excluding outliers because nothing had gone inherently wrong in those cases – the algorithm was the same. It is unfortunately the case that the optimisation can occasionally result in extreme validation error estimates for SVM models. We believed it would be wrong to ignore that fact and exclude those outliers as it would not reflect the nature of the surrogate model.

• We took the view that Winsorisation would retain those adverse results for SVM but make the statistical model manageable, whereas excluding the outliers would have a greater influence on the data distribution. Removing outliers is inherently asymmetric in its effect, whereas Winsorisation is symmetric by adjusting matching numbers of observations at opposite ends of the distribution, thereby preserving the median and interquartile range. Those outliers still have an effect, but their leverage is now limited.

• The table below shows the effect on validation RMSE of either removing the outliers or Winsorising the data compared to the original data. The extreme value for SD in the original data reveals the problem that needs to be addressed. The median remains the same with Winsorising, but removing outliers alters the IQR. The mean and SD change by a greater amount by removing outliers. Hence, Winsorising was an intervention that altered the data distribution to a lesser degree and thus led to our decision to prefer it over excluding outliers. 

 Original Data Outliers Removed Winsorised

Median 3.7535 3.7535 3.7535

IQR 1.6715 1.6258 1.6715

Mean 4.7895 4.1670 4.2204

SD 14.4098 1.6627 1.8235

Results

Line 385-386: Table 5 bad rendering. See suggestion for Table 1.

• Authors’ Response: Table 5 has been reformatted with a landscape page layout and a smaller font size. It is now Table 6 in the revised manuscript and can be found at line 428.

Discussion

Line 467-469: I am not sure of what you are asserting here. Which is the reason why you are relating sensor orientation to the external mechanical power computation?

• Authors’ Response: Further to our response on orientation in relation to one of your previous comments, the point we wished to make was that a single sensor’s kinematics must differ from the body CM’s kinematics. So it is impossible to determine the body’s mechanical power from the sensor’s inertial movements even if the change in orientation is known and considered. The peak power would apply to the sensor rather than the body. As we noted above, no single body-worn sensor can directly experience the kinematics of the body’s CM because it is held in a fixed position. Accordingly, we have provided more clarification over lines 509-516. 

Discretionary Revisions and typos

Material and Methods

Line 121-122: Maybe a picture of the setup would be beneficial to show the sensor attachment technique. I am saying that as it would be useful for the experimental setup repeatability.

• Authors’ Response: We have provided such a diagram to inform the reader, lines 155-160. See the new Fig1.tif. 

Results

Line 387-391: Maybe this paragraph has a more “methods” fashion.

• Authors’ Response: Since there were several stages to the procedures, the paragraph was written in this way to keep the reader on track so that they could make sense of the results presented. However, we have made amendments to make it more concise and feel more like a results section (lines 407-415). 

References

1. Monnet T, Decatoire A, Lacouture P. Comparison of algorithms to determine jump height and flight time from body mounted accelerometers. Sports Eng. 2014;17: 249–259. doi:10.1007/s12283-014-0155-1

2. Requena B, García I, Requena F, Saez-Saez de Villarreal E, Pääsuke M. Reliability and validity of a wireless microelectromechanicals based system (keimoveTM) for measuring vertical jumping performance. J Sports Sci Med. 2012;11: 115–122. 

3. Casartelli N, Müller R, Maffiuletti NA. Validity and Reliability of the Myotest Accelerometric System for the Assessment of Vertical Jump Height: J Strength Cond Res. 2010;24: 3186–3193. doi:10.1519/JSC.0b013e3181d8595c

4. Cooper G, Sheret I, McMillian L, Siliverdis K, Sha N, Hodgins D, et al. Inertial sensor-based knee flexion/extension angle estimation. J Biomech. 2009;42: 2678–2685. doi:10.1016/j.jbiomech.2009.08.004

5. Godwin A, Agnew M, Stevenson J. Accuracy of Inertial Motion Sensors in Static, Quasistatic, and Complex Dynamic Motion. J Biomech Eng. 2009;131: 114501. doi:10.1115/1.4000109

6. Luinge HJ, Veltink PH. Measuring orientation of human body segments using miniature gyroscopes and accelerometers. Med Biol Eng Comput. 2005;43: 273–282. doi:10.1007/BF02345966

---

## [Decision Letter · Decision Letter 1]

28 Jan 2022

Determining jumping performance from a single body-worn accelerometer using machine learning

PONE-D-21-30596R1

Dear Dr. White,

We’re pleased to inform you that your manuscript has been judged scientifically suitable for publication and will be formally accepted for publication once it meets all outstanding technical requirements.

Kind regards,

Chris Connaboy

Academic Editor

PLOS ONE

Additional Editor Comments (optional):

Reviewers' comments:

Reviewer's Responses to Questions

**Comments to the Author**

1. If the authors have adequately addressed your comments raised in a previous round of review and you feel that this manuscript is now acceptable for publication, you may indicate that here to bypass the “Comments to the Author” section, enter your conflict of interest statement in the “Confidential to Editor” section, and submit your "Accept" recommendation.

Reviewer #1: All comments have been addressed

Reviewer #2: All comments have been addressed

2. Is the manuscript technically sound, and do the data support the conclusions?

Reviewer #1: Yes

Reviewer #2: Yes

3. Has the statistical analysis been performed appropriately and rigorously? 

Reviewer #1: Yes

Reviewer #2: Yes

4. Have the authors made all data underlying the findings in their manuscript fully available?

Reviewer #1: Yes

Reviewer #2: Yes

5. Is the manuscript presented in an intelligible fashion and written in standard English?

Reviewer #1: Yes

Reviewer #2: Yes

6. Review Comments to the Author

Reviewer #1: (No Response)

Reviewer #2: I compliment the authors for a thorough and convincing revision. I apologise for my delay in providing you with my review. I did not understand a crucial point at first read of your answers to reviewers and then had a long time of no work in which I failed to comunicate with the journal.

7. PLOS authors have the option to publish the peer review history of their article (what does this mean?). If published, this will include your full peer review and any attached files.

Reviewer #1: **Yes: **Gavin L. Moir

Reviewer #2: No

---

## [Editor Report · Acceptance letter]

3 Feb 2022

PONE-D-21-30596R1 

Determining jumping performance from a single body-worn accelerometer using machine learning 

Dear Dr. White:

I'm pleased to inform you that your manuscript has been deemed suitable for publication in PLOS ONE. Congratulations! Your manuscript is now with our production department. 

Kind regards, 

on behalf of

Dr. Chris Connaboy 

Academic Editor

PLOS ONE